# Management of Malpositioned Double-Lumen Tubes: A Simulation-Based Training Exercise for Anesthesiology Residents

Bryan Hierlmeier *, Anand Prem, Suwarna Anand, Anna Lerant and Galina Ostrovsky

Department of Anesthesiology, The University of Mississippi Medical Center, Jackson, MS 39216, USA; aprem@umc.edu (A.P.); sanand2@umc.edu (S.A.); alerant@umc.edu (A.L.); gostrovsky@umc.edu (G.O.)
* Correspondence: bhierlmeier@umc.edu

**Abstract: Objectives:** Demonstrate the feasibility and effectiveness of using the AirSim® Bronchi airway simulator to teach anesthesia residents how to successfully troubleshoot and manage malpositioned double-lumen endotracheal tubes used for single-lung ventilation. **Design:** Prospective observational study. **Setting:** Simulation lab in a university-based anesthesiology residency training program. **Participants:** CA1 (PGY2) anesthesiology residents. **Interventions:** Each resident was initially tasked with troubleshooting a malpositioned double-lumen tube (DLT) on an AirSim® Bronchi (Trucorp, Craigavon, UK) airway simulator in one of the three positions and was timed on their performance. This first simulation was followed by completion of a survey that assessed the resident's level of confidence in repositioning a malpositioned DLT. Following the initial simulation, a didactic presentation was given to the residents on the management of malpositioned DLTs using a protocol, followed by a practice session. Two months later, each resident repeated the simulation exercise. A follow-up survey was conducted after each simulation, assessing the quality of the curriculum and subsequent level of confidence in performing the same tasks using a five-point Likert scale. **Measurements and Main Results:** Ten residents at the University of Mississippi Medical Center completed the simulation exercises and curriculum. The average time it took to troubleshoot the malpositioned DLT during the first simulation was 139 s, with only 30% of the residents correctly identifying the specific malposition and 40% correctly repositioning the DLT after initial malposition. The repeat simulation after two months showed significant improvement in efficiency, with the average time to complete the task decreasing from 139 s to 56 s. During the second simulation exercise, all 10 residents were able to correctly identify the DLT malposition and correctly reposition the DLT to the correct position. Additionally, residents' confidence levels in managing a malpositioned DLT improved significantly. Initially, 70% of the residents reported a lack of confidence in identifying and correcting a malpositioned DLT, but after the didactic curriculum and simulation training, 100% of the residents reported confidence in completing the task. **Conclusions:** The AirSim® Bronchi (Trucorp, Craigavon, UK) simulator can be used to effectively teach and evaluate residents on correctly identifying and repositioning a malpositioned DLT. The residents' performance and level of confidence were evaluated before and after the simulation curriculum. The results reveal that simulation-based training is an effective educational tool for improving clinical performance and confidence in identifying and appropriately managing a malpositioned DLT.

**Keywords:** anesthesia residents; simulation; lung isolation; double lumen tube

## 1. Introduction

Simulation has become a widely used training tool in anesthesiology residency programs, serving as an effective method to learn, improve, and evaluate clinical skills. Simulation-based learning allows for the opportunity to practice critical clinical skills and address any weaknesses within a safe learning environment that does not pose a risk

of endangering a patient. For cases that may have limited incidence or for techniques that are not routinely practiced, simulation can serve as a continuous learning tool to maintain necessary clinical skills. Learners and instructors express high levels of satisfaction with simulation as both an educational tool and a method of confidence-building that can improve clinical practice. Simulation training provides learners with the confidence to manage similar real-life scenarios, and this confidence can directly improve competence. By improving clinical knowledge, skills, and confidence, simulation-based training can result in safer and more efficient patient care [1,2].

The use of simulation in anesthesiology residency programs prepares residents for effective airway management and ventilation. To provide selective single-lung ventilation, a double-lumen tube (DLT) is the most preferred and commonly used device [3]. A DLT is an endotracheal tube used in tracheal intubation and designed to isolate the lungs anatomically and physiologically by selectively ventilating either the right or left lung. One-lung ventilation (OLV), while the other lung is collapsed, facilitates surgical exposure for thoracic operations [3]. Surgical indications for OLV include thoracic cases such as lung resection, video-assisted thoracoscopic surgery, lung transplantation, thoracic diaphragmatic hernia repair, and pleurodesis or pleurectomy [4]. OLV is also indicated in minimally invasive cardiac surgery, pericardiectomy, surgeries involving the thoracic aorta, thoracic sympathectomy, and the anterior approach to the thoracic spine [4]. Knowing how to both properly position a DLT and correct a malpositioned DLT is significant for airway management during thoracic surgery and in other cases where single-lung isolation is necessary.

Patients undergoing thoracic anesthesia can be at high risk due to a decreased respiratory reserve and the need for lung isolation. Gaining proficiency in a procedure that involves a learning curve can be very difficult to achieve in clinical practice given the patient population and limited exposure. Training in DLT placement and management is critical, as poor positioning can potentially lead to life-threatening consequences [3,5]. The malpositioning of a DLT can result in severely impaired ventilation, which can cause hypoxia, gas trapping, atelectasis, tension pneumothorax, cross-contamination and infections, and interference with surgical procedures [3–7]. It has been documented that between 9 and 28% of patients undergoing OLV during thoracic surgery develop severe arterial hypoxemia, with DLT malpositioning being a major contributing factor [5]. Studies have revealed that anesthesiologists who do not have routine thoracic anesthesia experience or training often incorrectly place DLTs and fail to manage malpositioned DLTs appropriately [4,8–10]. In approximately one-third of cases on average, anesthesiologists with limited training do not achieve the proper positioning of a left-sided DLT [8–10]. Even if initial DLT placement is performed successfully, (DLT) malpositioning commonly occurs, as blind intubation and lateral positioning of the patient can often displace the DLT [5]. After assessing with fiberoptic bronchoscopy, studies have revealed that DLTs become malpositioned in 25 to 54% of cases, requiring prompt repositioning [5,6]. Anesthesiologists should be comfortable in properly identifying and correcting a DLT malposition in a timely fashion to prevent potentially dangerous complications.

The purpose of this study is to demonstrate the feasibility of simulation-based training for effectively troubleshooting and managing malpositioned DLTs. This type of training allows exposure to the potential complications of one-lung ventilation and the management of a malpositioned DLT in a safe learning environment. Simulation, in particular, provides the hand–eye coordination training required for relatively difficult and infrequently performed procedures, such as DLT placement. It also familiarizes one with the equipment, such as the fiberoptic bronchoscope and the double-lumen tube, and its use in advanced airway management. We present the use of the "AirSim® Bronchi (Trucorp, Craigavon, UK)" for the structured training of anesthesia residents in the management of the one-lung ventilation needed for thoracic cases and other surgeries requiring lung isolation. The use of the "AirSim Bronchi Simulator" among residents has been documented to improve confidence in knowing how to successfully place a left double-lumen endotracheal

tube [10]. In this study, we propose that the use of the "AirSim Bronchi Simulator" can be extended to train residents in the management and correction of malpositioned DLTs, an incident that often occurs after initial placement. The high incidence of mismanaged DLTs among anesthesia providers, along with the increased risk of negative outcomes that result from malpositioned DLTs, emphasizes the importance of proper education and training in DLT management.

## 2. Methods

The AirSim® Bronchi (Trucorp, Craigavon, UK) airway simulators are life-sized, plastic upper-airway models that include the bronchial tree and reflect human anatomy relatively realistically, especially compared to other upper-airway simulators. Computerized tomography (CT) scanning was used to reveal that the fidelity of the AirSim® Bronchi (Trucorp, Craigavon, UK) simulator seems to be much higher based on objective anatomical measurements than the other simulators tested [11]. After an exempt status was granted by the IRB, all 10 CA1 anesthesiology residents at the University of Mississippi Medical Center were asked to participate in the study. At the time of the study, all CA1 residents had logged an average of three cases in which a DLT was used. The simulation was set up using the AirSim® Bronchi (Trucorp, Craigavon, UK) for lung isolation (Figure 1). A blue drape was placed over the bronchial anatomy to blind the residents from being able to visualize where the DLT was positioned as well as the light from the fiberoptic scope. A left-sided 35F DLT was malpositioned in one of the three positions shown in Figure 2. Simulation position B is when the DLT is left mainstem, position C had the bronchial cuff herniated above the carina, while position D was right mainstem. Upon entering the simulation room, the resident was instructed that soon after taking over a surgical case with an already inserted and positioned DLT, the surgeon complained that lung isolation was suboptimal. The resident was given a task to assess the DLT for correct positioning. If the DLT was identified as malpositioned, the resident would be required to correctly reposition the DLT (Figure 3). The timer was started when the fiberoptic bronchoscope (FOB) entered the DLT and was stopped upon removal of the FOB from the DLT. After completion of the simulation, the residents responded to a survey. Each resident was isolated as to not share simulation information with those who had not yet participated. After all 10 residents completed the simulation, a presentation was given by an anesthesiologist who regularly practices thoracic anesthesia. The presentation included a DLT malposition assessment protocol (Table 1) and simulation training on how to troubleshoot a malpositioned DLT. After the presentation, all 10 residents were able to practice management of DLT positioning using the simulator and the recently taught protocol. Two months after the original simulation, each of the 10 CA1 residents was brought back to the simulation center to repeat the original timed study and survey following completion. At the two-month mark, each of the CA1 residents had logged at least one case in which a DLT was used.

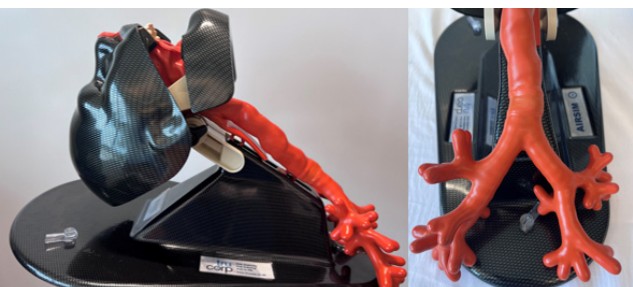

**Figure 1.** AirSim® Bronchi (Trucorp, Craigavon, UK).

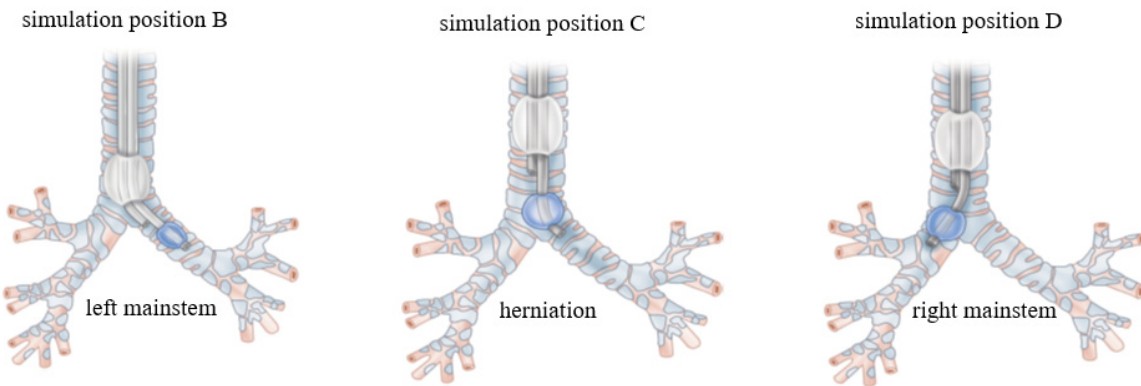

**Figure 2.** Malpositions of a left sided DLT [12].

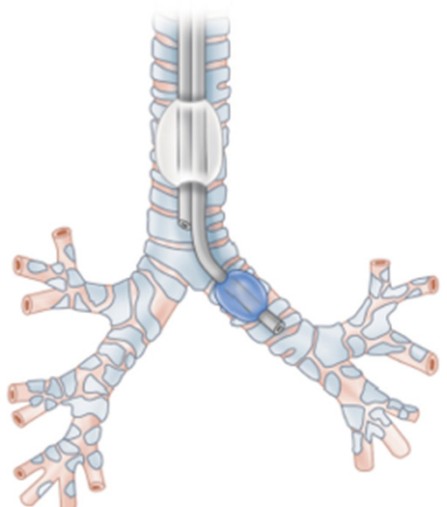

**Figure 3.** Properly placed left sided DLT [12].

**Table 1.** DLT malposition assessment protocol.

| Steps to Assess DLT Position |
| --- |
| Assess the tube length at the teeth: Is it too deep or too shallow? |
| Fiberoptic scope down the tracheal lumen to identify position; if unsure, go to step 3. |
| Fiberoptic scope down the bronchial lumen; if unsure, go to step 4. |
| With the scope through the bronchial lumen, deflate both the tracheal and bronchial cuffs and start withdrawing the DLT until you can confirm carina with left and right mainstem along with right-upper lobe for confirmation. |
| Once carina is accurately identified, (then) with the scope still through the bronchial lumen, advance the DLT into the correct position over the scope. |
| Confirm correct position of the fiberoptic scope through the tracheal lumen and visualize the bronchial lumen going into the left mainstem along with RUL identification. Inflate the tracheal, then the bronchial cuffs. |

## 3. Results

Ten residents completed the simulation assessment and curriculum. On the first attempt, only 30% of the residents could identify the specific type of DLT malposition and only 40% could correctly reposition the DLT to optimize lung isolation (after malposition). The average time it took the residents to correct the different malpositions B-D (Figure 2) on the first attempt is shown in Table 2. Position D seemed to be the most challenging for the residents to identify the malposition, and also consumed the most time, taking on average

156 s to evaluate and correct the DLT position. Resident ratings of self-confidence were reported after the completion of the first simulation via a survey (Table 3). After the first assessment, but before the presentation, 70% of CA1 residents stated that they were not confident in identifying and repositioning a malpositioned DLT.

**Table 2.** Time to identify and reposition the DLT.

| Simulation Position | First Attempt (Seconds) | Repeat Weeks Later (Seconds) | Difference (Seconds) |
|---|---|---|---|
| B | 134 | 34 | 99 |
| C | 126 | 56 | 70 |
| D | 156 | 76 | 80 |
| Average | 139 | 56 | 83 |

**Table 3.** Resident comfort in identification and positioning a malposition.

| Questionnaire Item | After First Assessment | After Curriculum + Simulation Training |
|---|---|---|
| Extremely comfortable | 0% | 10% |
| Very comfortable | 0% | 40% |
| Comfortable | 10% | 50% |
| Not comfortable | 70% | 0% |
| Very uncomfortable | 20% | 0% |

When comparing the first simulation with the second months later, the time it took to correctly identify the malposition and reposition the DLT drastically improved with all DLT malpositions, as seen in Table 2. On repeat simulation two months later, 100% of CA1 residents stated that they were confident in identifying a malpositioned DLT and correctly repositioning it after the simulation, with 40% being very comfortable and 10% being extremely comfortable.

## 4. Discussion

This study demonstrates the utility of simulators for the training and assessment of critical skills required in the management of malpositioned DLTs. It also has the potential advantage of interactive discussion between the expert and the trainee without compromising patient safety.

It reinforces the benefits of simulation-based training in anesthesiology residency in improving residents' clinical skills and confidence levels, especially in cases requiring the management of malpositioned DLTs. Most thoracic interventions require DLTs for OLV, but incorrect positioning or displacement of the DLT occurs at a significant rate and can lead to critical complications. These complications include hypoxia, atelectasis, high airway pressures, an accumulation of secretions, and a high incidence of infections after surgery [3–7]. Knowing how to promptly identify and reposition a malpositioned DLT is a critical clinical skill for anesthesiologists. Our residents found that the AirSim® Bronchi (Trucorp, Craigavon, UK) is a highly effective tool for learning and practicing the management of patients needing one-lung ventilation with a DLT.

In our study, the specific type of malposition was identified by only 30% of residents, while 40% of the residents could properly reposition the DLT prior to simulation training. Initially, residents took 139 s on average to identify and correct the malpositioned DLT. Most of the residents (70%) responded in the initial survey that they were not comfortable identifying and repositioning a malpositioned DLT. With limited to no exposure and training regarding DLT positioning, the residents were initially not properly equipped to

manage a patient with a DLT. However, following this initial assessment, the residents were trained using a DLT malposition assessment protocol presentation provided by an anesthesiologist with expertise in thoracic cases, in addition to further training with the simulator. This curriculum paired with simulation training proved to greatly improve residents' skill levels and confidence in DLT management. In a follow-up survey (Table 3) after the simulation training, 100% of residents responded that they are at least comfortable with identifying and repositioning a malpositioned DLT, with 40% being very comfortable and 10% being extremely comfortable. Additionally, the average time it took for the residents to correct the malpositioned DLT significantly reduced from 139 seconds in the initial assessment to 56 seconds after simulation training. The life-threatening complications that can arise from DLT malpositioning are often time-sensitive, and there is a critical need to respond quickly. The significant improvement in skill and response time for correcting a malpositioned DLT reveals the benefit of using simulation to train residents to provide the safest and most effective patient care.

Some inherent limitations are apparent in this study design as simulation cannot exactly replicate a clinical environment with actual patients. Although the AirSim® Bronchi (Trucorp, Craigavon, UK) serves as a highly accurate representation of human airway anatomy, real-life clinical scenarios using a DLT may differ from simulation, as patients can have complex airways with various pathologies. The pressure of managing a patient in a stressful clinical environment, such as the operating room, can also affect performance in a way that cannot be assessed accurately in a simulation. Additional limitations to this study include the lack of a control group and a formal assessment tool to determine whether simulation training had a direct impact on performance in a clinical setting. Training should not be solely limited to simulation, as residents should have ample opportunities to manage patients in the clinical setting under a diverse set of circumstances to build up their skill level. Despite these limitations, there are major benefits to simulation-based training that the clinical environment cannot always provide. Protecting patients' safety and well-being while also ensuring that physicians acquire the necessary clinical skills can be difficult to simultaneously achieve solely in the clinical environment. Simulation training can be the answer to supplement and amplify medical education, as learners can dedicate more time to mastering clinical skills in a safe environment without potentially harming a patient.

## 5. Conclusions

In conclusion, the results of our study further emphasize the benefit of simulation-based training as an educational tool to teach and practice the management of malpositioned DLTs during OLV. This study demonstrated that a curriculum for teaching management of malpositioned DLTs with a high-fidelity simulator, such as the AirSim® Bronchi (Trucorp, Craigavon, UK), is both feasible and invaluable to enhancing clinical skills and confidence among residents with limited experience in thoracic anesthesia. Time is of the essence, particularly when dealing with patients with significant comorbidities.

**Author Contributions:** Resources, A.L.; Writing—original draft, B.H. and G.O.; Writing—review & editing, A.P. and S.A. All authors have read and agreed to the published version of the manuscript.

**Funding:** This research received no external funding.

**Institutional Review Board Statement:** Ethical review and approval were waived for this study as it is already part of a training curriculum for the anesthesia residency. Patient consent was waived due to this being part of the training curriculum.

**Informed Consent Statement:** Patient consent was waived due to activity being part of the training program.

**Conflicts of Interest:** The authors declare no conflict of interest.

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
