# Peer review of "Management of Malpositioned Double-Lumen Tubes: A Simulation-Based Training Exercise for Anesthesiology Residents"

_ime, doi:10.3390/ime1010003_

Round 1

Reviewer 1 Report

This is an interesting study about simulation-based training program to assess the effectiveness of the AirSim Bronchi airway simulator. The testing subjects were the anesthesia residents and  the task was to place the double-lumen endobronchial tubes and  to troubleshoot the malposition. It is important to use simulation in training infrequently encountered clinical procedures and scenarios and the results of this study showed promising effects.

Comments:

Point-1: The study subjects were a single group of participants from one institution. Please describe the recruitment process in more details.

Point-2: The assessments of the training curriculum were based on the residents’ survey of their confidence and also the pretest and posttest performance from the simulation. Please describe who were those assessors of the tests in this study.

Point-3: It would be helpful to know whether these participants had any further opportunity of clinically practicing the DLEB tube procedure before the posttest follow-up assessment was conducted. In addition, any data from milestone or EPA to correlate the performance of these 10 residents?

Point-4: The DLEB tube malposition assessment protocol serves as an important tool for assessment of the competency and proficiency of the performance of the residents.  Please discuss a little bit more about the validity and reliability of such test instrument.

Point-5 (page 3, line 109): The authors cited the reference-11 to support AirSim Bronchi simulator as a valid simulator. However, the data from the reference-11 (also the data from Schebesta et al., Degrees of reality: airway anatomy of high-fidelity human patient simulators and airway trainers. ANESTHESIOLOGY. 2012;116:1204–9) did not indicate the fidelity of such airway simulator for “placement and correction” of malpositioned DLEB tube. Please discuss a little more on which anatomic dimensions of the AirSIm Bronchi are relevant to the scenario of placing DLEB tube for novice trainers.

Point-6 (page 3, line 117): Past tense for “.. the surgeon complained ….was....”

Point-7 (page 3, line 144): “Figure 1” should be “Figure 2”.

Point-8 (page 3, line 115): The quality of the Figures 2 and 3 needs to be improved. 2B: “to far” should be “too far”? 2C should be the “herniation” scenario and 2D should be the “right mainstem” positon?

Point-9 (page 4, line 139): The residents were required to identify “LUL” instead of “RUL”?

Point-10 (page 5, line 145): Please quantify the data in the Table-2. Any statistical significance of the difference? What is the distribution of the time spending on identifying the malposition and the time spending on correctly placing the malpositioned tube in each test case?

Point-11 (Page 5, line 151): The simulation test was repeated two months later. Any reason for the “two” months?

Ponit-12 (page 6, line 189): About the time to identify and reposition the tube, are there any quantitative description of the time (e.g., mean SE, n =10) instead of only mentioning 139 s and 56 s, respectively?

Point-13 (page 6 line 199): About the limitation of using simulator for teaching, here is the right place to cite appropriate simulation-based clinical training literature to discuss and compare (e.g., references 9 and 10).  Also, the authors need to compare and discussion the results from the references with those from the present study.

Point-14: The authors cite the references form internet sources (e.g., StatPearls in the references 1,2, and 5). It would be more convincing if some references from rather formal academic journals been cited to use as support or argument. 

Author Response

Point-1: The study subjects were a single group of participants from one institution. Please describe the recruitment process in more details.

  • All CA1 residents (10) were asked to participate.

Point-2: The assessments of the training curriculum were based on the residents’ survey of their confidence and also the pretest and posttest performance from the simulation. Please describe who were those assessors of the tests in this study.

            It was the same attending for all who had extensive experience in thoracic anesthesia.

Point-3: It would be helpful to know whether these participants had any further opportunity of clinically practicing the DLEB tube procedure before the posttest follow-up assessment was conducted. In addition, any data from milestone or EPA to correlate the performance of these 10 residents?

  • Each resident did log at least on clinical case in which a DLT was used. At the time of the study, we did not use EPAs

Point-4: The DLEB tube malposition assessment protocol serves as an important tool for assessment of the competency and proficiency of the performance of the residents.  Please discuss a little bit more about the validity and reliability of such test instrument.

  • This is the protocol that is clinically used by our thoracic anesthesiologist at the University of Mississippi Medical Center

Point-5 (page 3, line 109): The authors cited the reference-11 to support AirSim Bronchi simulator as a valid simulator. However, the data from the reference-11 (also the data from Schebesta et al., Degrees of reality: airway anatomy of high-fidelity human patient simulators and airway trainers. ANESTHESIOLOGY. 2012;116:1204–9) did not indicate the fidelity of such airway simulator for “placement and correction” of malpositioned DLEB tube. Please discuss a little more on which anatomic dimensions of the AirSIm Bronchi are relevant to the scenario of placing DLEB tube for novice trainers.

  • This is the only airway simulator we have to place lung isolation devices as it shows accurate airway anatomy with the trifurcation of the right upper lobe as well as the secondary carina at the end of the left main bronchus.

Point-6 (page 3, line 117): Past tense for “.. the surgeon complained ….was....”

            Corrected

Point-7 (page 3, line 144): “Figure 1” should be “Figure 2”.

            Corrected

Point-8 (page 3, line 115): The quality of the Figures 2 and 3 needs to be improved. 2B: “to far” should be “too far”? 2C should be the “herniation” scenario and 2D should be the “right mainstem” positon?

            Correct

Point-9 (page 4, line 139): The residents were required to identify “LUL” instead of “RUL”?

  • RUL not LUL

Point-10 (page 5, line 145): Please quantify the data in the Table-2. Any statistical significance of the difference? What is the distribution of the time spending on identifying the malposition and the time spending on correctly placing the malpositioned tube in each test case?

  • We took only averages and did not differentiate the time between identification and reposition.

Point-11 (Page 5, line 151): The simulation test was repeated two months later. Any reason for the “two” months?

 That was the time it took for all of the CA1s to log at least 1 DLT case in the OR

Ponit-12 (page 6, line 189): About the time to identify and reposition the tube, are there any quantitative description of the time (e.g., mean SE, n =10) instead of only mentioning 139 s and 56 s, respectively?

            We only took averages

Point-13 (page 6 line 199): About the limitation of using simulator for teaching, here is the right place to cite appropriate simulation-based clinical training literature to discuss and compare (e.g., references 9 and 10).  Also, the authors need to compare and discussion the results from the references with those from the present study.

We are unable to compare the studies referenced as those were dealing with initial placement of a DLT and not the identification a malposition.

Point-14: The authors cite the references form internet sources (e.g., StatPearls in the references 1,2, and 5). It would be more convincing if some references from rather formal academic journals been cited to use as support or argument.

These references were found through pubmed

Reviewer 2 Report

A nice study and well explained study design. Methodology is fine, and results seem to be appropriately derived from the data presented. 

Several things in the paper that need to be addressed:

1) Figure 2 - Are the pictures presented for Sim C and Sim D flip-flopped with their labels? It seems like the one that says the tube is down the R side is actually down the L side and/or herniated? and the one where the cuff is supposed to be herniated is actually down the R side?  Also - it seems like this figure is using pictures and diagrams from another source. These should probably be re-crafted to be original to the authors of this work, at least in some way. Or, if that's not possible, then the reference for the diagrams should be summarized into one figure legend and not all over the page. 

2) Table 1 - the protocol - there is no mention of what the operator is supposed to do with inflation/deflation of the various cuffs at each step. this needs to be included because otherwise the reader assumes the entire protocol is happening while all cuffs are fully inflated as in the diagrams. 

3) Table 2 needs a legend and the columns also need to indicate units. I am assuming the "134" is referring to "seconds" but this needs to be part of the table. Tables need to be able to be viewed separate from text and still have a reader know exactly what is being presented. 

4) Tabe 3 also needs a legend and labels similar to point #3 above. 

Author Response

  • Figure 2 - Are the pictures presented for Sim C and Sim D flip-flopped with their labels? It seems like the one that says the tube is down the R side is actually down the L side and/or herniated? and the one where the cuff is supposed to be herniated is actually down the R side?  Also - it seems like this figure is using pictures and diagrams from another source. These should probably be re-crafted to be original to the authors of this work, at least in some way. Or, if that's not possible, then the reference for the diagrams should be summarized into one figure legend and not all over the page. 

Corrected.

  • Table 1 - the protocol - there is no mention of what the operator is supposed to do with inflation/deflation of the various cuffs at each step. this needs to be included because otherwise the reader assumes the entire protocol is happening while all cuffs are fully inflated as in the diagrams. 

Added to table 1

  • Table 2 needs a legend and the columns also need to indicate units. I am assuming the "134" is referring to "seconds" but this needs to be part of the table. Tables need to be able to be viewed separate from text and still have a reader know exactly what is being presented. 

Added

  • Tabe 3 also needs a legend and labels similar to point #3 above. 
    1. added

Reviewer 3 Report

Major Strengths- This study does a great job of showing the value of simulation in training for dangerous issues that arise regularly in practice that are hard to difficult to train providers how to resolve due to the requirement to resolve the issue quickly rather than slowly allowing for education and understanding. Additionally, the access to these cases on a regular basis is also limited, so simulation is the ideal way to ensure all trainees have access to experiencing and evaluating a number of possible scenarios, and knowledge of and ability to practice a concise and effective protocolized solution. 

Weakness- No control group to identify if senior residents gain this knowledge over time without simulation. However, this is stated in the discussion and I don't feel it would have been in the scope of this study to attempt to establish that control.

Specific Suggestions-

Abstract:

- Line 24. Please specify if the residents weren't able to identify if the DLT was malpositioned or if they weren't able to identify the specific type of malposition (this comes up again later in the manuscript). It is confusing that 30% were able to "identify the malposition", but 40% were able to fix the problem. I assume they were not able to identify the specific malposition, but it should be clearly stated.

Introduction:

- Line 94. You may want to add (Company Name, City, State, Country) after the simulator name here when you reference it for the first time.

Methods:

- Line 113: the word "about" in this sentence should be replaced with "able".

- Line 115: please define the positions in the text. Referring to the figure is appropriate, but they should be defined at least once in the text.

Results:

- Line 142: the language "exact type" is used here. This language or something similar should be used in the abstract, and the "types" should be spelled out in the text as stated above.

- Line 153: You state that 100% of the residents were confident in identifying a malpositioned tube, but you never state how many of them identified the correct type of malposition. Was this data collected or analyzed?

- There does not appear to be any statistical analysis described or p-values presented. Where there any T-Tests completed between the pre and post identification, correction, or timing data? It's not mandatory, however, if it was completed it would add even more power to this impactful work.

Discussion:

- Line 175: Please see comment on clarifying what the 30% value in this sentence actually identifies.

- Line 178: This sentence contains the word "are" which should likely be changed to "were" to allow for correct tense requirements in the sentence.

Author Response

- Line 24. Please specify if the residents weren't able to identify if the DLT was malpositioned or if they weren't able to identify the specific type of malposition (this comes up again later in the manuscript). It is confusing that 30% were able to "identify the malposition", but 40% were able to fix the problem. I assume they were not able to identify the specific malposition, but it should be clearly stated.

            Corrected

Introduction:

- Line 94. You may want to add (Company Name, City, State, Country) after the simulator name here when you reference it for the first time.

Methods:

- Line 113: the word "about" in this sentence should be replaced with "able".

            Corrected

- Line 115: please define the positions in the text. Referring to the figure is appropriate, but they should be defined at least once in the text.

            Added

Results:

- Line 142: the language "exact type" is used here. This language or something similar should be used in the abstract, and the "types" should be spelled out in the text as stated above.

             Corrected

- Line 153: You state that 100% of the residents were confident in identifying a malpositioned tube, but you never state how many of them identified the correct type of malposition. Was this data collected or analyzed?

- There does not appear to be any statistical analysis described or p-values presented. Where there any T-Tests completed between the pre and post identification, correction, or timing data? It's not mandatory, however, if it was completed it would add even more power to this impactful work.

            This was not done

Discussion:

- Line 175: Please see comment on clarifying what the 30% value in this sentence actually identifies.

            Corrected

- Line 178: This sentence contains the word "are" which should likely be changed to "were" to allow for correct tense requirements in the sentence. Corrected

Reviewer 4 Report

The authors describe the use of a simulation based exercise to teach Anesthesiology residents troubleshooting and repositioning malpositioned DLTs.

The paper needs to be overall needs to be formatted in syntax and grammar to be read easier.  There are numerous paragraphs that seem very redundant in the message they convey to the reader.  The writing seems very basic and needs to be fixed to be published in a scientific journal.  There are odd capitalizations throughout the manuscript that do not make sense.

In the authors, the final author's credentials are listed as "and MS".  This needs to be fixed. 

Introduction:

Lines 40-51 needs more references to support the statements made.

line 66-the "morbidity" associated with thoracic anesthesia needs to be described better.  

lines 66-86: this whole paragraph read poorly and needs to be re formatted.

line 94 : "AirSim Bronchi Simulator" should be "AirSim Bronchi (trucorp, N. Ireland).  This needs to be standardized in the manuscript everywhere it is referenced.

Methods:

Questions to be answered:

-how far along were the CA1's in their training?  Was this at the beginning of their CA1 year, or the end?  How familiar are they with double lumen tubes, yet alone single lumen ETTs?

-have the CA1's had a thoracic rotation already?  Did they complete a rotation in between part I and part II of the test?

Results and Discussion:

-using an N of 10 students, there are broad statements made on the utility of simulation and the ability to have improvement in training.  Without further delineating the residents' preexisting knowledge and training, I find it rather brash to make such statements.  The authors also state that they had no formal assessment tool to determine whether it would affect their clinical performance.

The manuscript, in my opinion, needs a complete reworking in grammar and syntax, and the conclusions drawn need to be reframed.

Author Response

In the authors, the final author's credentials are listed as "and MS".  This needs to be fixed. 

Corrected 

Introduction:

Lines 40-51 needs more references to support the statements made.

line 66-the "morbidity" associated with thoracic anesthesia needs to be described better.  Re-worded 

lines 66-86: this whole paragraph read poorly and needs to be re formatted. -

 - removed a few sentences. 

line 94 : "AirSim Bronchi Simulator" should be "AirSim Bronchi (trucorp, N. Ireland).  This needs to be standardized in the manuscript everywhere it is referenced.      Corrected 

Methods:

Questions to be answered:

-how far along were the CA1's in their training?  Was this at the beginning of their CA1 year, or the end?  How familiar are they with double lumen tubes, yet alone single lumen ETTs?  This was about 8 month into training. each had only a few DLT cases 

-have the CA1's had a thoracic rotation already?  Did they complete a rotation in between part I and part II of the test? No 

Round 2

Reviewer 1 Report

It seems that all the points I made have been responded and corrected.

Reviewer 4 Report

-Having an MS as the senior author rather than an MD is rather odd.  It would be better to have an MD as a senior author, and if need be, have the MS as the first author to give the paper more crediblity

-trucorp, N. Ireland needs to be stated only the first time.  Not after every time the Air-Sim Bronchi device is named.

-to be more valuable, I think more students than a N=10 would make the study have more credence.